Taxonomic evaluation of selected Ganoderma species and database sequence validation

Jargalmaa Suldbold 1
Eimes John A. 2
Park Myung Soo 1
Park Jae Young 1
Oh Seung-Yoon 1
Lim Young Woon ywlim@snu.ac.kr 1
1 School of Biological Sciences, Seoul National University , Seoul , South Korea
2 University College, Sungkyunkwan University , Suwon , South Korea
Dentinger Bryn
Electronic publication date: 2017 Jul 27
Publication date: 2017
Volume: 5
Electronic Location ID: e3596
Received 2017 Feb 18; Accepted 2017 Jun 29
Copyright: ©2017 Jargalmaa et al.
Copyright year: 2017
Copyright holder: Jargalmaa et al.
License: This is an open access article distributed under the terms of the Creative Commons Attribution License, which permits unrestricted use, distribution, reproduction and adaptation in any medium and for any purpose provided that it is properly attributed. For attribution, the original author(s), title, publication source (PeerJ) and either DOI or URL of the article must be cited.
License URL: https://creativecommons.org/licenses/by/4.0/

Keywords: Ganoderma, Polypores, Medicinal fungi, Genbank sequence validation

Funding: National Institute of Biological Resources under the Ministry of Environment NIBR2016-53 This project was supported by the National Institute of Biological Resources under the Ministry of Environment (Project No.: NIBR2016-53), Republic of Korea. The funders had no role in study design, data collection and analysis, decision to publish, or preparation of the manuscript.

==============================
Species in the genus Ganoderma include several ecologically important and pathogenic fungal species whose medicinal and economic value is substantial. Due to the highly similar morphological features within the Ganoderma, identification of species has relied heavily on DNA sequencing using BLAST searches, which are only reliable if the GenBank submissions are accurately labeled. In this study, we examined 113 specimens collected from 1969 to 2016 from various regions in Korea using morphological features and multigene analysis (internal transcribed spacer, translation elongation factor 1-α, and the second largest subunit of RNA polymerase II). These specimens were identified as four Ganoderma species: G. sichuanense, G. cf. adspersum, G. cf. applanatum, and G. cf. gibbosum. With the exception of G. sichuanense, these species were difficult to distinguish based solely on morphological features. However, phylogenetic analysis at three different loci yielded concordant phylogenetic information, and supported the four species distinctions with high bootstrap support. A survey of over 600 Ganoderma sequences available on GenBank revealed that 65% of sequences were either misidentified or ambiguously labeled. Here, we suggest corrected annotations for GenBank sequences based on our phylogenetic validation and provide updated global distribution patterns for these Ganoderma species.

Introduction

Fungal species in the genus Ganoderma Karst. (Ganodermataceae, Polyporales) include ecologically important wood decay fungi of which some species are a well-known component of traditional Asian medicine. Several species of Ganoderma have been reported to cause diseases associated with trees, including basal stem rot disease in oil palm caused by G. boninense (Susanto, Sudharto & Purba, 2005), and root-rot disease of Acacia trees caused by G. steyaertanum, G. mastoporum, and G. philippii (Glen et al., 2009). Despite the pathogenic nature of many Ganoderma species, many species, especially taxa identified as G. lucidum in Asia, are believed to possess medicinal characteristics, and have been used in traditional Asian medicine for millennia (Bishop et al., 2015). Ganoderma byproducts are increasingly being used in western medicine and related health industries, and the most recent estimate of the annual economic value of Ganoderma byproducts (calculated in 1995) was ∼1.6 billion USD (Chang & Buswell, 1999); adjusting for inflation and economic growth, this annual economic value is likely several billion dollars in 2017. As our understanding of the biochemistry and genetics of Ganoderma biocompounds increases in tandem with increasing medicinal and economic demand for these byproducts, accurate identification of Ganoderma species is critical.

Ganoderma is likely a relatively young genus, originating in the tropics and recently expanding its range into temperate zones. The estimated number of Ganoderma species ranges from 250 to >400 (Moncalvo, Wang & Hseu, 1995; Richter et al., 2015). The morphology of Ganoderma species is characterized by a crusty or shiny pileus surface and a two-layered basidiospore wall with a truncated apex. Due to the high similarity of basidiocarp features, it is likely that the Ganoderma is the most difficult genus to accurately identify to species of all polypores (Ryvarden & Gilbertson, 1993). In fact, the Ganoderma have been described as being in a state of “taxonomic chaos” (Ryvarden, 1991). Indeed, the wide range of estimates for the number of Ganoderma species exemplifies the ambiguity that permeates the taxonomy of this genus. Early efforts to apply molecular markers toward the resolution of Ganoderma taxonomy used sequences from internal transcribed spacer (ITS) and partial large subunit rDNA (Moncalvo et al., 1994; Moncalvo, Wang & Hseu, 1995; Moncalvo, 2000) and nearly complete small subunit rDNA sequences (Hong & Jung, 2004; Douanla-Meli & Langer, 2009). More recently, Wang et al. (2012) assessed the identification of European G. lucidum (a species originally described from England) and East Asia G. lucidum using three loci and determined that the East Asia samples were genetically distinct from their European counterparts and conspecific with G. sichuanense. Further complicating matters, Cao, Wu & Dai (2012) combined morphological characters with phylogenetic analyses of six loci and proposed naming East Asian G. lucidum as G. lingzhi; however, G. lingzhi is regarded as a synonym of G. sichuanense, the name proposed in 1983 (Wang et al., 2012). Following the rules of fungal nomenclature, the name G. sichuanense should be given preference over any synonyms (Richter et al., 2015).

The dramatic increase in available DNA sequences from molecular phylogenetic studies has helped resolve the taxonomy of numerous fungal groups. Sequences from the ITS have been particularly useful in the recent development of DNA barcoding of the fungi (Schoch et al., 2012), although other loci such as translation elongation factor 1-α (tef1-α) and the second largest subunit of RNA polymerase II (rpb2) (Liu, Whelen & Hall, 1999; Matheny et al., 2007) have been instrumental in resolving ambiguous evolutionary relationships among Ganoderma species. While molecular advances have in general led to improvements in phylogenetics and taxonomy, the relative ease and speed of DNA barcoding also has the potential to increase the confusion often associated with fungal taxonomy. Up to 20% of designated sequences in public databases may be erroneous because of improper species identification (Bridge et al., 2003; Vilgalys, 2003; Nilsson et al., 2006). This problem is acute within the genus Ganoderma due to the absence of reliable morphological characteristics, a high rate of synonymous classification, and incorrect taxonomic assignments in public databases. The combination of morphology and molecular analysis (ideally multi-locus) can resolve the issues associated with this growing problem of misidentified sequence in public databases (Jung et al., 2014).

Sequencing costs have recently fallen precipitously as has the speed and ease with which specimens can be identified via sequencing. As a result, an increasing number of non-specialists, including edible mushroom cultivators as well as collectors and re-sellers of medicinal fungi, have relied on commercial sequencing companies to identify specimens. These identifications are often made using BLAST searches, which are only reliable if the GenBank submissions are accurately labeled. Thus, a re-evaluation of Ganoderma species that are endemic to Korea is needed and the accuracy of sequence databases such as GenBank should be investigated.

In Korea, the first report of Ganoderma lucidum (reported as Fomes japonicus) was 1934 (Murata, 1934). The classification system of Ganoderma proposed by Imazeki (1952) divided the Ganoderma into two subgenera, with laccate species (including G. lucidum) in the subgenus Ganoderma and non-laccate species (including E. applanatum) in the subgenus Elfvingia. To date, five Ganoderma species, G. applanatum, G. lipsiense, G. lucidum G. neojaponicum, and G. tsugae, have been recorded in Korea (Kaburagi, 1940; Lee, 1981; Kang, 1991; Kwon et al., 2016). G. lipsiense, however, is synonymous with G. applanatum (Moncalvo & Ryvarden, 1997), therefore only four Ganoderma species were known in Korea. Importantly, most Korean Ganoderma species were reported solely on basidiocarp morphology without detailed descriptions or molecular data.

In this study, our main objective was to build a Korean Ganoderma inventory using morphology and molecular analysis and to correct species misidentifications in GenBank. In addition, we provide updated global distribution patterns of these four Ganoderma species using newly validated GenBank sequences.

Table 1 Representative Ganoderma specimens from the Seoul National University Fungus Collection (SFC) used in this study.

Species	Specimen no.	Collection sites	Accession number	
		Locality	Latitude/Longitude	ITS	rpb2	tef1-α	
G. sichuanense	SFC20120721-08	Gimpo-si, Gyeonggi-do	37°36′08.43″N/126°46′33.51″E	KY364244			
	SFC20150624-06	Pohang-si, Gyeongsangbuk-do	36°04′20.02″N/129°12′36.56″E	KY364245	KY393267	KY393279	
	SFC20150630-14	Jongno-gu, Seoul	37°34′28.50″N/126°59′38.92″E	KY364246	KY393268	KY393280	
	SFC20150812-48	Jongno-gu, Seoul	37°34′28.30″N/126°59′44.28″E	KY364247		KY393281	
	SFC20150918-07	Jongno-gu, Seoul	37°34′26.70″N/126°59′36.12″E	KY364248	KY393269	KY393282	
	SFC20160315-03	Yangyang-gun, Gangwon-do	38°07′17.46″N/128°33′06.78″E	KY364249		KY393283	
	SFC20160420-01	–a		KY364250			
G. cf. adspersum	SFC20141001-16	Inje-gun, Gangwon-do	37°57′11.80″N/128°19′24.52″E	KY364251	KY393270	KY393284	
	SFC20141001-22	Inje-gun, Gangwon-do	37°57′02.55″N/128°19′29.46″E	KY364252	KY393271	KY393285	
	SFC20140701-31	Inje-gun, Gangwon-do	37°56′50.06″N/128°19′47.85″E	KY364253			
	SFC20160115-20	Yangpyeong-gun, Gyeonggi-do	37°29′20.09″N/127°36′34.14″E	KY364254	KY393272	KY393286	
G. cf. applanatum	SFC20141001-24	Inje-gun, Gangwon-do	37°56′46.46″N/128°20′00.98″E	KY364255	KY393273	KY393287	
	SFC20141001-25	Inje-gun, Gangwon-do	37°57′13.39″N/128°19′16.80″E	KY364256			
	SFC20141012-02	Inje-gun, Gangwon-do	37°57′12.84″N/128°19′18.18″E	KY364257			
	SFC20150930-02	Inje-gun, Gangwon-do	38°07′30.12″N/128°12′10.08″E	KY364258	KY393274	KY393288	
G. cf. gibbosum	SFC20130404-21	Sangju-si, Gyeongsangbuk-do	36°31′47.93″N/128°04′27.06″E	KY364259			
	SFC20140702-12	Seogwipo-si, Jeju-do	33°14′54.58″N/126°21′03.42″E	KY364260	KY393275		
	SFC20140703-17	Jeju-si, Jeju-do	33°26′24.63″N/126°37′39.49″E	KY364261			
	SFC20150418-05	Gwanak-gu, Seoul	37°27′21.15″N/126°56′57.18″E	KY364262			
	SFC20150612-11	Donghae-si, Gangwon-do	37°27′51.09″N/129°01′00.16″E	KY364263			
	SFC20150630-23	Jongno-gu, Seoul	37°34′24.07″N/126°59′36.17″E	KY364264	KY393276	KY393289	
	SFC20150701-06	Jeju-si, Jeju-do	33°19′31.62″N/126°16′50.62″E	KY364265			
	SFC20150723-01	Jongno-gu, Seoul	37°34′22.66″N/126°59′37.49″E	KY364266			
	SFC20150812-02	Jongno-gu, Seoul	37°34′21.62″N/126°59′38.18″E	KY364267			
	SFC20150812-35	Jongno-gu, Seoul	37°34′20.40″N/126°59′45.21″E	KY364268			
	SFC20150812-36	Jongno-gu, Seoul	37°34′27.81″N/126°59′44.59″E	KY364269			
	SFC20150918-03	Jongno-gu, Seoul	37°34′24.63″N/126°59′39.11″E	KY364270	KY393277	KY393290	
	SFC20150918-08	Jongno-gu, Seoul	37°34′20.23″N/126°59′44.05″E	KY364271	KY393278	KY393291	
	SFC20160713-09	Jeju-si, Jeju-do	33°29′26.67″N/126°36′08.87″E	KY364272			
Notes.

a No information.

Materials and Methods

Sample collection and morphological analysis

Specimens were collected from 1969 to 2016 from various regions in Korea and stored at the Seoul National University Fungus Collection (SFC) and Korea Mushroom Resource Bank (KMRB). These specimens were initially identified as G. lucidum, G. neojaponicum, and G. applanatum (Table S1). Because the complex characteristics of the basidiocarps of Ganoderma have contributed to confusion in the taxonomy of this genus, we sorted specimens using macro- and micro-morphological observations (Gilbertson & Ryvarden, 1986; Cao, Wu & Dai, 2012). Initially, three morphological features were observed for 113 specimens: shape of basidiocarp, pore number per mm at hymenophore, and basidiospore size. Pore number was calculated as the mean of five 1 mm transects across hymenium. In order to observe basidiospores, slide preparations mounted in 3% KOH were made from dried tissue for each specimen and examined with a Nikon 80i light microscope (Nikon, Tokyo, Japan).

DNA extraction, amplification, and sequencing

A total of 29 recently collected Korean specimens were chosen for DNA sequencing (Table 1). Genomic DNA was extracted using a modified CTAB extraction protocol (Rogers & Bendich, 1994). The ITS region, partial tef1-α, and partial rpb2 regions were amplified using the primers ITS1F/ITS4b (Gardes & Bruns, 1993), EF1-983F/EF1-2218R (Rehner & Buckley, 2005), and fRPB2-5F/bRPB2-7R2 (Liu, Whelen & Hall, 1999; Matheny et al., 2007), respectively. PCRs were performed on a C1000TM thermal cycler (Bio-Rad, Richmond, CA, USA) using AccuPower PCR premix (Bioneer Co., Daejeon, Korea) according to the methods described in Park et al. (2013). PCR products were electrophoresed through a 1% agarose gel stained with EcoDye DNA staining solution (SolGent Co., Daejeon, Korea) and purified using the Expin™ PCR Purification Kit (GeneAll Biotechnology, Seoul, Korea) according to the manufacturer’s instructions. DNA sequencing was performed at Macrogen (Seoul, Korea) using an ABI3700 automated DNA sequencer. Sequences obtained from specimens were proofread using chromatograms in MEGA v. 6 (Tamura et al., 2013).

Phylogenetic analysis

Phylogenetic analysis was carried out in two steps. First, phylogenetic trees using ITS, tef1-α, and rpb2 sequences were constructed using only specimens of Korean Ganoderma species. Second, we downloaded all Ganoderma sequences obtained from the search query “Ganoderma” in GenBank. SFC amplicon sequences of ITS, tef1-α, and rpb2 were aligned with Ganoderma sequences downloaded from GenBank using the default settings of MAFFT v.7 (Katoh & Standley, 2013). Maximum likelihood (ML) trees were constructed with RAxML 8.0.2 (Stamatakis, 2014) using the GTRGAMMA model of evolution and 1,000 bootstrap replicates. Coriolopsis cf. caperata was used as an outgroup for all three phylogenetic trees (Binder et al., 2013).

Validation and distribution of GenBank Ganoderma sequences

We used BLAST to validate Ganoderma sequences in GenBank. GenBank sequences for G. sichuanense were available for all three loci (ITS, tef1-α, and rpb2); however, there were insufficient sequences for the other species at the tef1-α and rpb2 loci. Thus, our validation for these species is limited to the ITS locus. ITS sequences from each Korean species were used for the BLAST searches and sequences were downloaded based on similarity and coverage. We downloaded all sequences that had a similarity of >90% at the ITS. We excluded short sequences by removing those that had coverage of <50%. Neighbor Joining (NJ) analyses were performed using these sequences to determine the correct sequence identity for each Ganoderma species. NJ trees were constructed with MEGA v. 6, using the Kimura 2-parameter model and 1,000 bootstrap replicates. All work with GenBank was performed on September 20, 2016.

We used the validated sequence information of four Ganoderma species to generate a map of the global distribution. Distribution information of each species was extracted from published papers and direct GenBank submissions.

Figure 1 Basidiocarps of Ganoderma; G. sichuanense (A–B), G. cf. adspersum (C–D), G. cf. applanatum (E–F), and G. cf. gibbosum (G–H).

Scale bars: (A–H) = 1 cm.

Figure 2 (A) Box plot representing pore number per mm of four Ganoderma species: G. sichuanense (Gsi), G. cf. adspersum (Gad), G. cf. applanatum (Gap), and G. cf. gibbosum (Ggi). Boxes represent the interquartile range between first quartile and third quartile. Bold line in the box is the median and filled circles represent individual outlying data points. (B) Scatter plot of basidiospore size among the four species (mm).

Four samples were observed for each species.

Results

Evaluation of Ganoderma specimens based on morphological and molecular analyses

All 113 specimens identified as Ganoderma were used in the preliminary portion of this study. These samples were reexamined based on distinguishable morphological characters. First, specimens with laccate basidiocarps and long stipes were distinguished from other specimens with non-laccate basidiocarps. Laccate specimens initially identified as G. lucidum, G. lingzhi and G. neojaponicum were identified as G. sichuanense using molecular analysis based on ITS, tef-1, and rpb2. Basidiocarps were reniform to circular with long subcylindrical stipe (Figs. 1A and 1B), had circular or angular pores that were at a density of 5–6 per mm (Fig. 2A), and basidiospore size was (9.7) 10.4–11.1 (12.2) × (6.4) 6.6–6.9 (7.4) μm (Fig. 2B).

While all non-laccate specimens were similar to G. applanatum, they could be separated into three different morphology types. Type A specimens had sessile basidiocarps, and were attached directly to the tree at its base with no stipe (Figs. 1C and 1D). Pore number was 3–4 per mm (Fig. 2A) and basidiospore size range was (7.8) 8.3–10.6 (11.3) × (5.1) 5.4–7.4 (7.8) μm (Fig. 2B). Type B specimens had sessile basidiocarps with no stipes (Figs. 1E and 1F), a pore number range of 5–7 per mm (Fig. 2A), and basidiospore size of (8.0) 8.1–8.5 (8.9) × (5.3) 5.4–5.8 (6.3) μm (Fig. 2B). The basidiocarps of type C specimens were attached to broad-leaved tree stumps with short stipes (Figs. 1G and 1H), a pore number range of 4–5 per mm (Fig. 2A), and basidiospore size of (7.7) 8.5–9.2 (9.4) × (4.9) 5.6–6.0 (6.5) μm (Fig. 2B).

The ITS region was successfully amplified and sequenced for 29 representative specimens. However, sequences for the tef1-α and rpb2 regions were obtained from fewer specimens (Table 1). Phylogenetic relationships inferred from the ITS, tef1-α, and rpb2 ML trees exhibited a clear distinction between the four species (Fig. 3). This phylogeny supported the identification of the laccate specimens as G. sichuanense. The three morphological types of non-laccate specimens clearly separated into three species. We re-named type A Ganoderma cf. adspersum, type B Ganoderma cf. applanatum, and type C Ganoderma cf. gibbosum because type specimens were not included in this study. Within the ITS, tef1-α, and rpb2 phylogenies, G. cf. adspersum formed a supported clade with G. cf. gibbosum. G. cf. applanatum formed a distinct cluster with these two species to the exclusion of G. sichuanense (Fig. 3).

Figure 3 Phylogenetic tree for Ganoderma and related species based on a maximum likelihood (ML) analysis of the internal transcribed spacer (ITS), the second largest subunit of RNA polymerase II (rpb2), and translation elongation factor 1-α (tef1-α).

ML trees were constructed with RAxML 8.0.2 using the GTRGAMMA model of evolution and 1,000 bootstrap replicates. Bootstrap scores of >50 are presented at the nodes. Branches that involved SFC sequences are in bold. The scale bar indicates the number of nucleotide substitutions per site.

GenBank sequence validation and distribution of four Ganoderma species

Using BLAST searches and phylogenetic analysis, we were able to validate the sequences of four Ganoderma species in GenBank. Of 249 ITS sequences that were identified by this study as G. sichuanense, 239 were annotated in GenBank as G. sichuanense, G. lucidum or G. lingzhi, seven were undetermined (Ganoderma sp.) and three were mislabeled (two as G. tsugae and one as G. luteomarginatum). One GenBank sequence submission was incorrectly identified as G. lingzhi and three were misidentified as G. sichuanense. 111 GenBank submissions that were initially annotated in GenBank as G. lucidum were neither Asian (G. sichuanense) or European (G. lucidum) Ganoderma species. Thus, of 354 GenBank submissions labeled as G. sichuanense or its synonyms, 115 (32%) were found to belong to different species (Fig. 4, Table S2).

Figure 4 Incorrect names applied to Ganoderma sequences in GenBank.

Color-coded taxon identifiers indicate initial GenBank annotations (number of sequences are in parentheses); “other” indicates annotations labeled as “Ganoderma clone” or as non-Ganoderma genera. Numbers in barred circles represent GenBank submissions that were incorrectly identified as indicated species.

88 ITS sequences were defined by this study as G. cf. adspersum. Sixteen G. cf. adspersum sequences were mislabeled in GenBank as either G. applanatum (4) or G. australe (12). Seven were ambiguously labeled (as Ganoderma sp.). One sequence was labeled “Ganoderma clone” and four were misidentified as non-Ganoderma genera. In addition, we found four GenBank sequences that were erroneously labeled as G. adspersum (Fig. 4, Table S2).

Of 85 ITS sequences defined as G. cf. applanatum in this study, just 46 (54%) were correctly labeled as G. applanatum or its synonym, G. lipsiense. Four ITS sequences were mislabeled Ganoderma species (three as G. adspersum and one as G. oregonense) and seven were ambiguously labeled (e.g., Ganoderma sp.). Of 70 GenBank sequence submissions originally labeled as G. applanatum or its synonym G. lipsiense, 24 (34%) were found to belong to different species (Fig. 4, Table S2).

We identified 34 GenBank sequences as G. cf. gibbosum. A total of 17 GenBank sequences were correctly annotated, while 14 were initially misidentified as other Ganoderma species (seven as G. applanatum, five as G. australe, one as G. fulvellum, and one as G. lucidum). Three G. cf. gibbosum sequences were ambiguously labeled as “Ganoderma sp.”. A total of 14 of 31 (45%) GenBank sequence submissions that were initially identified as G. gibbosum were found to be different species (Fig. 4).

Based on the corrected database, we generated a distribution map for each species (G. sichuanense, G. cf. adspersum, G. cf. applanatum, and G. cf. gibbosum) (Fig. S1). G. sichuanense was distributed throughout Asia (e.g., China, Japan, Korea, Bangladesh, Malaysia, and Nepal); although nearly 70% of the G. sichuanense sequences were described from China. While some sequences were identified as from Poland and Italy, their specimen information lacked confirmation due to directly deposition without publication. Most G. cf. adspersum sequences were from European countries (Italy, Germany, Poland, United Kingdom, Austria, Finland, and France), while a small number of sequences were from Asia (India, Japan, and Korea). G. cf. applanatum sequences had a global distribution (USA, Canada, Lithuania, Hungary, Germany, Poland, Korea, and Antarctica). G. cf. gibbosum sequences were mostly limited to Asia (Korea, China, Japan, and India), with one group of sequences identified as from Poland (unpublished sequences).

Discussion

Morphological and molecular analysis of Korean Ganoderma specimens collected during the last fifty years indicated that there are four Ganoderma species in Korea. Among the four previously described Ganoderma species, G. neojaponicum and G. tsugae were not found in this study. Although one specimen that was identified as G. neojaponicum was shown to be G. sichuanense, further study is needed to establish whether G. neojaponicum and G. tsugae exist in Korea.

G. sichuanense was previously identified and named G. lucidum in Korea. Recently Kwon et al. (2016) suggested that G. lucidum cultivated in Korea (locally known as Yeongji) was actually G. lingzhi, although we argue for renaming all G. lingzhi as G. sichuanense (see above). G. sichuanense is easily distinguished from the other three Korean Ganoderma species by differences in surface texture of the pileus and basidiospore size (Moncalvo & Ryvarden, 1997; Tham, 1998). Laccate pileus with longer stipe and larger basidiospores than other Asian Ganoderma species are typical characters of G. sichuanense (Wang et al., 2012). Furthermore, phylogenetic analysis confirmed that the Korean G. sichuanense sequences used in this study were nearly identical to the epitype for G. sichuanense (KC662402) (Yao, Wang & Wang, 2013).

The three species with non-laccate basidiocarps, G. cf. adspersum, G. cf. applanatum, and G. cf. gibbosum, have similar morphological characteristics which often lead to misidentification of these species, although basidiocarp morphology has been suggested to differentiate these species. Basidiocarps of G. cf. adspersum (40–100 mm) are usually thicker than those of G. cf. applanatum (20–60 mm) at the base. In addition, the undersides of the basidiocarps of G. cf. adspersum have a decurrent attachment, whereas those of G. cf. applanatum tend to emerge sharply at right angles from the host stem (Ryvarden & Gilbertson, 1993; Schwarze & Ferner, 2003). In a radial section of the hymenophore of the older parts of the fruiting body, those of G. cf. adspersum remain empty but the pores of G. cf. applanatum become filled with a white mycelium (Breitenbach & Kränzlin, 1986). G. cf. adspersum is distinguished from G. cf. applanatum by having larger basidiospores (Steyaert, 1972; Ryvarden & Gilbertson, 1993). G. cf. gibbosum is distinguished from G. cf. applanatum by the presence of the stipe (Blume & Nees von Esenbeck, 1826). It has been suggested, however, that stipe formation may be an adaptive feature because individuals of the G. applanatum-australes complex can develop a stipe in the tropics and stipe formation can be induced in the laboratory in strains of G. applanatum-australes complex species (Moncalvo & Ryvarden, 1997). Nevertheless, Korean specimens in our study were distinguished by three characteristics: The presence of the stipe discriminated G. cf. gibbosum from G. cf. adspersum and G. cf. applanatum (Fig. 1) and larger basidiospore and pore size discriminated G. cf. adspersum from G. cf. applanatum (Fig. 2).

Despite similar morphology, a multigene phylogenetic analysis showed that G. cf. adspersum, G. cf. applanatum, and G. cf. gibbosum, are distinct species (Fig. 3) corresponding to clades 2, 1, and 5, respectively, of the Ganoderma global phylogeny that was constructed by Moncalvo & Buchanan (2008). Our results also support previous phylogenetic reconstructions where G. cf. adspersum and G. cf. applanatum were clearly separated by rDNA analysis and further distinguished by species specific PCR primers (Gottlieb, Ferrer & Wright, 2000; Guglielmo et al., 2008). While G. cf. adspersum and G. cf. gibbosum formed a supported clade separate from G. sichuanense, G. cf. adspersum and G. cf. gibbosum appear to be closely related with 100% bootstrap support in tef1-α and rpb2 phylogenetic trees (Fig. 3).

Our study found that the number of misidentified sequences of the four Ganoderma species in GenBank was substantial (Table S2, Fig. 4), with ITS sequences being significantly more likely to be misidentified than other loci due to their over-representation among phylogenetic markers. Open DNA databases (DB) such as GenBank are an important tool for species identification. In medicinal fungi, such as Ganoderma species, the need for satisfactory taxonomic sampling and accurate identification in DBs is critical. Among the four species, the highest number of ITS sequences listed on GenBank was those of G. sichuanense (Fig. 4) and the unusually high number of G. sichuanense sequences found on GenBank is likely due to the economic and medicinal importance of the species. In order to minimize confusion and misidentification in future studies, we strongly recommend that the names G. lucidum and G. lingzhi be avoided and all new submissions of this species be labeled G. sichuanense. The distribution of G. sichuanense appears to be limited to Asia, with specimens reported from China, Korea, Japan, Bangladesh, Malaysia, and Nepal (Fig. S1).

Our study shows that G. cf. adspersum sequences in GenBank were commonly misidentified as G. australe (Fig. 4). Based on morphological analysis, G. cf. adspersum was considered a synonym of G. australe by Ryvarden (1976) and Ryvarden & Gilbertson (1993); however, Smith & Sivasithamparam (2000), using ITS sequence data, argued that G. adspersum and G. australe are two distinct species. G. adspersum was commonly reported from Europe, where type specimens were collected (Moncalvo & Ryvarden, 1997; Smith & Sivasithamparam, 2000). Our analysis confirmed that G. cf. adspersum occurs in Europe, but is distributed in Asia and North America as well (Fig. S1). Erroneously annotated sequences were also common among G. cf. applanatum GenBank submissions. 34% of sequences that were initially annotated as G. cf. applanatum were shown to be other species while just 54% of authentic G. cf. applanatum sequences were initially annotated as such in GenBank. A total of 13 sequences that were labeled as G. lipsiense in GenBank were included in the G. cf. applanatum clade because G. lipsiense is synonymous with G. cf. applanatum (Moncalvo & Ryvarden, 1997). Our analysis confirmed that G. cf. applanatum has a global distribution (Moncalvo & Ryvarden, 1997) with sequences reported from Europe, Asia and North America (Fig. S1). Nearly half of all GenBank sequences annotated as G. cf. gibbosum were misidentified and the same proportion of authentic G. cf. gibbosum sequences were initially annotated as different species. G. cf. gibbosum has a primarily Asian distribution, and the ecto-type was initially reported from Java, Indonesia (Blume & Nees von Esenbeck, 1826), although scattered samples were reported from Eastern Europe.

In conclusion, as we constructed phylogenetic trees using reference sequences from GenBank, it became apparent that many Ganoderma reference sequences were misidentified. In this study, we identified incorrectly labeled sequences on GenBank and constructed new phylogenies with reference sequences that were correctly assigned to specific taxa. This study will provide a framework for future efforts to replace inaccurate public information with reliable taxonomic assignments. We strongly encourage the authors of previously submitted specimens that have been shown to be misidentified or use inappropriate species names (i.e., lucidum and lingzhi) to correct these submissions on GenBank. This improvement is vital not only for fungal taxonomists, but given the diverse ecological, medicinal, and economic impacts of Ganoderma species, this project will be of value to researchers across multiple disciplines.

Supplemental Information

Figure S1 Supplementary Figure 1

Click here for additional data file.

Table S1 Supplementary Table 1

Click here for additional data file.

Table S2 Supplementary Table 2

Click here for additional data file.

We greatly appreciate two anonymous reviewers for their detailed and kind comments.

Additional Information and Declarations

Competing Interests

Author Contributions

Data Availability

The authors declare there are no competing interests.

Suldbold Jargalmaa conceived and designed the experiments, performed the experiments, analyzed the data, wrote the paper, prepared figures and/or tables, reviewed drafts of the paper.

John A. Eimes analyzed the data, wrote the paper, reviewed drafts of the paper.

Myung Soo Park conceived and designed the experiments, performed the experiments, analyzed the data, reviewed drafts of the paper.

Jae Young Park conceived and designed the experiments, analyzed the data, reviewed drafts of the paper.

Seung-Yoon Oh performed the experiments, analyzed the data, reviewed drafts of the paper.

Young Woon Lim conceived and designed the experiments, analyzed the data, contributed reagents/materials/analysis tools, wrote the paper, reviewed drafts of the paper.

The following information was supplied regarding data availability:

The raw data (DNA sequences) are available in GenBank: KY364244 –KY364272.

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
