# Peer review of "Taxonomic evaluation of selected Ganoderma species and database sequence validation"

_PeerJ, doi:10.7717/peerj.3596_

## Round 0.1 · original submission · Major Revisions

First, my apologies for the unusual delay in returning this decision. The reviewers were both very thorough and put in a lot of time and effort into this and I am very grateful for it.

Overall this is agreement that this is a useful contribution and I am very supportive of its publication. However, the reviewers were divided on what rank of revision to recommend and while it is clear that the vast majority of revisions necessary are minor, the taxonomic problem raised by Reviewer #2 is not. Given the importance of this particular issue, I adopted the Major category for this decision. That said, I do not think it will require a major effort to address, but rather that it is an important conclusion with potentially wide-reaching impacts and thus needs to be considered carefully.

In addition to the comments by the reviewers and references therein, I am attaching a PDF of an overview of Ganoderma systematics that I hope the authors will find useful and that should at the very least be cited.

Reviewer 1 ·

Basic reporting

This manuscript is well-written and appropriately structured. The Introduction covers the key issues relating to the problems of Ganoderma taxonomy and aims are clearly stated. However, the initial text L31-L56 relating to medicinal aspects (L31-L56) could be cut to a sentence or two since this is a phylogenetic paper with no specific focus on medicinal value of Ganoderma.

Figures and tables are all relevant and appropriate.
Fig. 4 legend could be clarified –this shows the extent of mis-naming on GenBank and should be titled “Incorrect names applied to Ganoderma. sequences in GenBank” or similar. Also recolour Fig4a to make the pie segments clearer and indicate clearly here which names are synonyms.
For Fig.2 state in legend or below titles on x-axis, the number of samples studied for each species. Table 1 should have lat/long location details –all from Korea so this can be stated in legend.

Abstract should include more specific detail of what was found –eg that XX-YY% of GenBank sequences are misidentified; also that the three loci yielded concordant phylogenetic information; and that morphological information was often not definitive in identifying species (but maybe addition of host tree info would be of additional help?

Data presented here are ITS/rpb2 or tef-1a sequences and all have GenBank numbers presented (but not accessible to me at present, presumably released is delayed until publication).

Experimental design

The level of taxon sampling for the four species examined in detail here is good (29 specimens) and salient samples have been studied at all three relevant loci. Phylogenetic analyses are conducted robustly and presented clearly.

A larger number of samples have been examined morphologically. It would be very useful to present some host tree (also whether dead or alive) information as this is useful in taxonomy of other Ganoderma species –this information could be added as a column in Table 1. There is brief mention of host tree (L188) but more could be made of this.

I liked Fig4. and the authors’ presentation of the number of mis-named specimens on GenBank

Validity of the findings

This paper adds clarity to Ganoderma taxonomy and especially to the problems with GenBank accessions. Some of these problems are due to the global “Ganoderma community” failing to agree on a name for G. sichuanense /lucidum /lingzhi. The authors’ phylogenetic analyses confirm that these for a single clade with no evident substructure. Do they feel bold enough to call for the epithets lucidum /lingzhi to be dropped? I am not a Ganoderma afficianado but such a removal of historical debris is nomenclature would be a good move!

At over 3 pages, the discussion should be rewritten more clearly and succinctly, and would benefit from subheadings each dealing with specific issues. Eg. G. sichuanense /lucidum /lingzhi; unreliability of some macromorphological characters; biogeographic factors; cleaning up of GenBank. With regard to the last suggested subheading, I’d suggest that the authors contact Conrad Schoch (Dr. Fungus at GenBank) to ask his advice as to how the errors on GenBank could be expunged.

Additional comments

A clear well-written paper. Needs some work on the discussion to more forcefully make the key points emerging from the authors’ work. I hope the authors are willing to be bold here!

·

Basic reporting

This paper is clearly and logically written. The research questions are well defined and meaningful. Past literature on the subject is well covered, with one notable exception (see below). However, several sentences / statements are imprecise and/or awkward, for instance:

L22-24: The species used in traditional medicine also decay wood. Therefore, change, e.g., "while others" to "of which some species."

L28: "Asian G. lucidum": change to, e.g., "taxa identified as G. lucidum in Asia"?

L32: Most, if not all taxa in these studies were wrongly identified as G. lucidum. Therefore, change "G. lucidum" to e.g. "Ganoderma species (most of which wrongly labeled G. lucidum)"

L82: Liu et al. 1999 and Matheny et al. 2007 did not study evolutionary relationships among Ganoderma species. Move these references after rpb2.

L87-90: There is a contradiction here. On one hand it is stated that there is an absence of reliable morphological characteristics (L87-88); on the other hand it is stated that morphological data are a vital component of species identification (L89-90).

L122-123: “These specimens were initially identified as G. lucidum, G. neojaponicum, and G. applanatum (Table S1).” Table S1 does not indicate the name of the specimens.

L127-128: “Pore number was calculated as the mean of five 1 mm transects across a basidiospore.” Should read "hymenium" rather than "basidiospore"?

L162: “ITS sequences from each species were used for the BLAST searches”. "Korean" should be added before "species", I think.

L195-196: "The three morphological types of G. applanatum clearly separated into species…" This is awkward. Change "G. applanatum" to e.g. "non-laccate specimens"

L248-249: "Two species, G. adspersum and G. gibbosum were newly described in Korea in this study." NO, these two species are not newly described in this study. At best, they are new record from Korea (but see below).

Experimental design

The data sampling and methods are appropriate to answer the research questions. The methods are generally described with sufficient details. There are a few shortcomings, however:

L144-145: “Sequences obtained from specimens were aligned and proofread using MEGA v. 6”. Does this mean that only the text files were used to edit the sequences and that the chromatograms were not examined? This needs to be double-checked with the authors: if this is the case, then sequence accuracy is questionable.

I am guessing that there were ambiguously aligned regions in ITS and also in EF1 introns. If so, were these regions trimmed before phylogenetic analyses?

L165: Details about the NJ settings should be given.

Validity of the findings

The findings are overall novel and valid. However, there are several shortcomings:

L177: “Laccate specimens initially identified as G. lucidum, G. lingzhi and G. neojaponicum were identified as G. sichuanense.” As it stands, it seems that the first three taxa were re-identified as G. sichuanense based on morphology only. Really? I am guessing that sequence data helped a lot and if so, this should be stated.

L204-205: G. lucidum IS NOT a synonym of G. sichuanense!!

Taxonomic issues:
The authors named the three morphological types of non-laccate specimens found in Korea as G. applanatum, G. adspersum, and G. gibbosum (L195-198). There is no solid basis for that since the respective type specimens were not examined morphologically neither sequenced. The authors comprehensively and extensively discuss the controversies and problems associated with these names, including the high number of misidentified sequences in GenBank. It is clear, however, that only type studies can solve this problem, as discussed in Moncalvo & Ryvarden (1997), Moncalvo (pp. 23-45 In Ganoderma Diseases of Perennial Crops, eds J. Flood, P.D. Bridge, and M. Holderness, CAB International), and briefly also in Moncalvo & Buchanan (Mycol. Res. 2008:425-436). The authors did not examine the latter paper, which reported a phylogeny of non-laccate Ganoderma (the G. applanatum-australe complex) from a worldwide sampling of 96 isolates; 8 clades representing 8 putative species were identified, but not named because type specimens were not included in the study. I have two recommendations:
(1) For the time being, in absence of comprehensive type studies, I suggest the three Korean species found in this study to be named G. cf. applanatum, G. cf. adspersum, and G. cf. gibbosum.
(2) The authors should try to place these three Korean taxa within the global phylogenetic framework in Moncalvo & Buchanan. Do they correspond to any of the 8 putative species identified in that study or do they potentially represent additional species?

---

## Round 0.2 · accepted · Accept

This manuscript is ready for publication, but one reviewer did have a useful suggestion for an alternative to fixing the taxonomic errors in GenBank:

"I am happy that the issues identified in the initial reviews have been carefully and fully addressed. The only suggestion I have is that the authors may be able to use the UNITE database/PlutoF platform to correct some of the taxonomic errors which they identified (https://plutof.ut.ee/#/unite/view/583563). As noted previously, modification of GenBank entries is difficult, whereas this is easier to do via UNITE. If they feel unable to actually modify entries etc via UNITE, they could suggest this possibility in their discussion."

The authors should seriously consider making these changes in UNITE, which would be a useful service for future Ganoderma research. If they do decide to provide this service to the research community, it would be important to include a statement mentioning this in the manuscript.

Reviewer 1 ·

Basic reporting

Happy that the authors have addressed the issues previously raised under this section

Experimental design

Happy that the authors have addressed the issues previously raised under this section

Validity of the findings

Happy that the authors have addressed the issues previously raised under this section

Additional comments

I am happy that the issues identified in the initial reviews have been carefully and fully addressed. The only suggestion I have is that the authors may be able to use the UNITE database/PlutoF platform to correct some of the taxonomic errors which they identified (https://plutof.ut.ee/#/unite/view/583563). As noted previously, modification of GenBank entries is difficult, whereas this is easier to do via UNITE. If they feel unable to actually modify entries etc via UNITE, they could suggest this possibility in their discussion.

·

Basic reporting

The authors have carefully and meaningfully addressed the comments and suggestions of the reviewers. In my opinion, the current ms should be accepted as is.

Experimental design

good

Validity of the findings

good